# Benefit of Optical Coherence Tomography–Angiography in Patients Undergoing Transsphenoidal Pituitary Adenoma Surgery: A Prospective Controlled Study

**DOI:** 10.3390/diagnostics14161747

**Published:** 2024-08-12

**Authors:** Elsa Toumi, Fabien Almairac, Lydiane Mondot, Albert Themelin, Anne-Gaëlle Decoux-Poullot, Philippe Paquis, Nicolas Chevalier, Stéphanie Baillif, Sacha Nahon-Esteve, Arnaud Martel

**Affiliations:** 1Ophthalmology Department, University Hospital of Nice, Université Côte d’Azur, 06000 Nice, France; baillif.s@chu-nice.fr (S.B.); nahon-esteve.s@chu-nice.fr (S.N.-E.); martel.a@chu-nice.fr (A.M.); 2Neurosurgery Department, University Hospital of Nice, Université Côte d’Azur, 06000 Nice, France; almairac.f@chu-nice.fr (F.A.); paquis.p@chu-nice.fr (P.P.); 3UR2CA PIN, Université Côte d’Azur, 06000 Nice, France; 4Radiology Department, University Hospital of Nice, Université Côte d’Azur, 06000 Nice, France; mondot.l@chu-nice.fr (L.M.); themelin.a@chu-nice.fr (A.T.); 5Endocrinology Department, University Hospital of Nice, Université Côte d’Azur, 06000 Nice, France; decoux-poullot.ag@chu-nice.fr (A.-G.D.-P.); chevalier.n@chu-nice.fr (N.C.)

**Keywords:** pituitary adenoma, vascular theory, mechanical theory, optic chiasm compression, OCT-angiography, vascular density

## Abstract

Background: Although visual field (VF) defects are common in compressive pituitary adenoma (CPA), their pathophysiology has not been fully elucidated. The mechanical theory (i.e., direct compression of the optic chiasm by the CPA) and the vascular theory (i.e., compression of the vessels supplying the visual path by the CPA) or their association could explain the visual impairment. The aim of this study was to determine whether the vascular density (VD) improved after surgical decompression of the optic chiasm in CPA patients and whether OCT-A could help to identify predictive factors for postoperative visual recovery. Methods: A prospective controlled study was conducted in patients who underwent transsphenoidal pituitary adenoma surgery. Patients were divided into two groups: with CPA and without CPA (NCPA). All patients underwent a neuro-ophthalmological examination, VF testing, macular and optic disc structural OCT [retinal nerve fiber layer (RNFL) and ganglion cell complex (GCC) thicknesses] and OCT-A before and then 1 and 6 months after surgery. Results: Twenty-four eyes and fourteen eyes were included, respectively, in the CPA and NCPA groups. None of the VD parameters assessed by OCT-A were significantly improved after surgery in the CPA group. In the CPA group, the mean macular superficial VD was significantly decreased at 6 months. The multivariate analysis failed to identify any preoperative parameters predictive of postoperative VF improvement. Conclusions: Our preliminary findings suggest that the visual impairment observed in CPA patients could not be explained by the vascular theory. None of the preoperative OCT-A parameters allowed a postoperative VF recovery assessment. Trial registration number NCT04074642, ID-RCB 2019-A01186-51 date of registration 30 July 2019.

## 1. Introduction

The optic chiasm is a critical neuroanatomical area corresponding to the decussation of a portion of the visual fibers. Previous studies have shown that the optic chiasm is mainly vascularized through its inferior part, the hypophyseal arteries being critical [1,2]. Laterally, the optic nerve is vascularized by the ophthalmic artery that is located about 5 mm in front of the hypophyseal arteries [1] (Figure A1).

Although VF defects are common in compressive pituitary adenoma (CPA), their exact pathophysiology has not been fully elucidated. The mechanical theory postulates that a compression of the optic chiasm is responsible for a retrograde axonal degeneration [3,4], visualized as a thinning of the macular ganglion cell complex (GCC) and retinal nerve fiber layer (RNFL) [5,6,7]. Although widely accepted in other optic nerve diseases such as glaucoma, the vascular theory has been poorly investigated in CPA [8,9,10,11,12]. A compression of the superior hypophyseal arteries and ophthalmic artery could explain the previously described VF defects and reduced macular and papillary vascular densities (VD) [13,14,15,16,17]. Finally, a combination of the mechanical and vascular theories in a variable portion could be responsible for the VF defects seen in CPA [18].

Several studies using OCT-Angiography (OCT-A) have shown that the macular and papillary VD were significantly reduced in CPA [13,14,15,19]. It is still unclear whether this vascular rarefaction is the consequence of a retinal GCC thinning due to reduced metabolic and vascular needs (mechanical theory) or results from reduced arterial blood flow caused by a vascular compression (vascular theory) (Figure A2).

Another critical but still unanswered challenge is the identification of a reliable preoperative parameter predictive of postoperative VF recovery [20]. Traditional spectral domain optical coherence tomography (SD-OCT) does not appear reliable enough to predict the postoperative visual outcomes. Several studies have suggested that OCT-A could be useful, but most of them had a retrospective design and did not include non-compressive pituitary adenoma (NCPA) patients as a control group [13,14,15,17].

The aims of this study were (i) to test the vascular theory by assessing whether the macular and papillary VD improved following optic chiasm decompression and (ii) to assess whether the preoperative macular and papillary VD could predict the postoperative VF improvement.

## 2. Materials and Methods

### 2.1. Study Design and Participants

A prospective controlled cohort study was conducted in the Ophthalmology, Neurosurgery, Endocrinology and Radiology departments of Nice University Hospital, France. 

All patients who underwent endoscopic transsphenoidal PA neurosurgery between January 2020 and January 2022 at Nice University Hospital were prospectively included. Patients were divided into 2 groups: with and without CPA (CPA and NCPA groups, respectively). Patients with a visual impairment were included in the CPA group. The other patients, operated on for endocrine reasons, were included in the NCPA group (control group).

Inclusion criteria were age > 18 years, diagnosis of PA confirmed on MRI, absence of any other cerebral disease and providing a signed written informed consent.

Non-inclusion criteria were ophthalmological diseases (amblyopia, high myopia, glaucoma and any other optic disc and macular disease), any neurosurgical disease (previous PA surgery, previous cerebral radiotherapy), low-quality OCT and OCT-A images, and pregnancy.

Exclusion criteria were withdrawal of the written informed consent and need for postoperative radiotherapy.

### 2.2. Data Recorded

Each patient underwent preoperative (1 month before surgery) and postoperative (1 month and 6 months after surgery) ophthalmological examinations. The best-corrected visual acuity (BCVA) was measured and recorded, and slit-lamp biomicroscopy, fundus examination, static automated perimetry (SAP), SD-OCT (GCC and RNFL thicknesses) and OCT-A findings were recorded. Endocrine venous samples were collected before and after surgery in all patients. 

### 2.3. Brain Imaging

All patients underwent brain MRI before and 6 months after surgery with high-resolution coronal T2-weighted and fat-suppressed T1-weighted sequences with gadolinium injection focused on the pituitary region. Manual segmentation of PA was performed by a single experienced neuroradiologist (LM), and the tumor volume was calculated in cube centimeters before and after surgery. 

### 2.4. Static Automated Perimetry

SAP was performed with the Humphrey Field Analyzer (Carl Zeiss Meditec, Jena, Germany) using the Swedish Interactive Threshold Algorithm Standard strategy, program 24-2. Mean deviation (MD) and Visual field index (VFI) values were used for the analysis, as well as the type of VF defect. Reliability criteria were fixation losses ≤ 20%, and false positive and false negative rates ≤ 33%.

### 2.5. Spectral-Domain Optical Coherence Tomography

SD-OCT (RTVue XR software, version 2017.1.0.151, Optovue Inc., Fremont, CA, USA) was used to examine the RNFL. The optic nerve head analysis measured the RNFL thickness calculated along a 3.45 mm diameter circle around the optic disc. SD-OCT (Spectralis Heidelberg Retina Angiograph + Optical Coherence Tomography, Heidelberg Engineering, Herthfordshire, UK) was used to examine the GCC thickness that was measured from the internal limiting membrane (ILM) to the outer boundary of the inner plexiform layer (IPL) in a 7 × 7 mm grid on the macula at the posterior pole. CPA duration was classified according to the following classification: acute compression (1–3 months) in case of VF impairment without GCC and RNFL alteration, subacute compression (3–6 months) in case of VF and GCC impairment but without RNFL impairment, and chronic compression (>6 months) in case of VF, GCC and RNFL alterations.

### 2.6. Optical Coherence Tomography–Angiography

OCT-A was performed with the RTVue XR Avanti (Visionix, Luneau technology operations, Chartres, France). We used 4.5 × 4.5 mm HD angio-disc and 6 × 6 mm angio-retina scans. AngioVue software (version 2017.1.0.151, Optovue Inc., Fremont, CA, USA) automatically provided VD values. The software (version 2017.1.0.151; Optovue, Inc., Fremont, CA, USA) automatically provided the VD in the whole en face macular area and papillary and peripapillary areas. 

AngioAnalytic™ software (version 2017.1.0.151, Optovue Inc., Fremont, CA, USA) automatically analyzed the VD in two different parafoveolar retinal vascular networks: the superficial capillary plexus (between the ILM and 9 μm above the junction between the IPL and the inner nuclear layer [INL]) and the deep capillary plexus (between 9 mm above the IPL–INL junction and 9 μm below the outer plexiform layer–outer nuclear layer junction). 

AngioAnalytic™ software automatically calculated the foveal avascular zone (FAZ) area in square millimeters in the 6 × 6 mm macular area in the full retinal plexus.

Papillary VD was measured in the whole en face papillary area, the peripapillary area (corresponding to a 0.75 mm wide annulus from the inner optic disc margin) and the inside disc area. Poor quality scans, defined as a Quality Index <7/10, off-range acquisitions, saccade or blinking artifacts, were excluded from the analysis. An improvement in macular or papillary VD was arbitrarily defined as a 10% increase in postoperative values from baseline.

### 2.7. Neurosurgery

The procedures were performed using a full right-nostril endoscopic transseptal approach. The nasal phase was performed using a 0° rigid endoscope (Storz^®^, Tuttlingen, Germany), and the sellar phase using a 0° and 30° rigid endoscope fixed on a dedicated support. Neuronavigation (Medtronic^®^, Dublin, Ireland) was systematically used. Large anterior sphenoidotomy was performed when feasible, and the bone fragment removed to access the intrasellar compartment was kept for subsequent closure. The sellar dura was opened, and the tumor was gently removed using conventional microsurgical methods. No excessive traction was applied on the optic chiasm. In case of invasion of the cavernous sinus, a tumor residue was deliberately left in place. A pathological examination was performed for all tumors. After resection, hemostasis was achieved after bipolar cauterization, compression with cottonoids and use of hemostatic products. An abdominal fat graft was systematically positioned in the sellar dura and fixed with the bone fragment using fibrinogen glue (Tisseel^®^, Baxter, Deerfield, IL, USA).

### 2.8. Primary Outcomes

To test the vascular theory, preoperative and postoperative macular and papillary VD were measured on OCT-A to assessed whether the VD was increased 6 months after surgery in CPA patients experiencing a VF improvement. 

VF improvement was defined as an improvement in the mean deviation of VF > 6 dB associated with a subjective improvement in VF scotoma assessed by two trained ophthalmologists (ET, AM). 

### 2.9. Statistics

Statistical analyses were performed using SPSS software (version 28; IBM, Armonk, NY, USA). Qualitative parameters are reported as frequencies and percentages, quantitative parameters as means (±SD). The normality of quantitative parameters was assessed using the Shapiro–Wilk test. Comparisons of patients’ groups according chiasma compression were performed using a Chi-squared test for qualitative parameters and an unpaired Student’s *t*-test, Welch’s *t*-test or Mann–Whitney U test depending on data distribution for quantitative parameters. The two-sided alpha risk was set at 5%. A binomial logistic regression was performed to assess the relationship between the VF improvement after surgery and the explanatory variables assessed in the bivariate analysis. Data multicollinearity was controlled using the Belsley–Kuh–Welsch technique. Heteroskedasticity and normality of residuals were assessed respectively using the White test and the Shapiro–Wilk test. The model performance was assessed using the Hosmer–Lemeshow test to confirm the goodness of fit of the logistic regressions. Results are presented as an odds ratio (OR) with the 95% confidence interval (CI).

## 3. Results

### 3.1. Patients’ Baseline Characteristics

Among the 38 patients operated on in the neurosurgery department of Nice University Hospital, 19 were excluded due to the poor quality of OCT-A images (*n* = 14), glaucoma (*n* = 2) and amblyopia (*n* = 3). The study flowchart is shown in Figure 1. Twenty-four eyes of 12 patients (63.16%) and fourteen eyes of 7 patients (36.84%) were included in the CPA and NCPA groups, respectively. Patients’ demographics are shown in Table 1. The mean PA volume was significantly larger in the CPA group compared to the NCPA group (10.65 cm^3^ and 3.77 cm^3^, respectively, (*p* < 0.01). The preoperative VF (*p* < 0.01), RNFL thickness (*p* < 0.01), GCC thickness (*p* < 0.01), whole disc VD (*p* < 0.01) and peripapillary VD (*p* < 0.01) on OCT-A were significantly lower in the CPA group (Figure A3 and Figure A4). No disc swelling was detected in both groups. 

### 3.2. Surgical Outcomes

The tumor size was significantly reduced after surgery in both groups.

No case of persistent chiasmatic compression was found on MRI 6 months after surgery. No patient required adjuvant radiotherapy. 

### 3.3. VF and SD-OCT Parameter Outcomes

There was no improvement in BCVA, VF mean deviation, mean RNFL thickness and mean GCC thickness 1 and 6 months after surgery in the NCPA group. There was no VF improvement in patients of the NCPA group (Table 2). 

In the CPA group, the BCVA was significantly improved 1 month (0.03 logMAR) and 6 months (0.04 logMAR) after surgery (*p* = 0.02). The VF mean deviation was also significantly improved 1 month and 6 months after surgery compared to baseline (−9.34 dB at baseline versus −3.35 dB at 1 month and −2.01 dB at 6 months; *p* < 0.01). Half of the patients (*n* = 6) experienced a VF improvement at 6 months. The GCC thickness (40.97 μm at baseline, 39.28 μm at 1 month and 38.01 μm at 6 months, *p* = 0.03) was significantly reduced 6 months after surgery but not the RNFL thickness.

### 3.4. OCT-A Parameter Outcomes

No significant change in OCT-A parameters observed 1 and 6 months after surgery in the NCPA group (Table 3).

In the CPA group, the mean macular sVD was significantly decreased at 6 months (46.18%) compared to baseline (48.66%, *p* = 0.02). To analyze the correlation between the VD parameters and the VF changes, the CPA group was divided into two subgroups: patients with and without VF improvement at 6 months. No significant improvement in macular and papillary VD was found in CPA patients who experienced a VF improvement. (Table 3 and Table A1 and Figure A5). 

### 3.5. Preoperative Prognostic Factors for Postoperative VF Improvement

A multivariate analysis was performed to determine whether the preoperative parameters could predict the postoperative VF improvement (Table 4). In the univariate analysis, the gender (*p* = 0.04) and the VF mean deviation (*p* = 0.04) were preoperative predictive factors for VF improvement, but these factors were no longer significant in the multivariate analysis. All the studied OCT-A parameters failed to predict the postoperative VF improvement. 

## 4. Discussion

In this prospective controlled study, we found that the macular and papillary VD did not improve following neurosurgical decompression of the optic chiasm, suggesting that the vascular theory could not be predominant in the visual impairment found in CPA. In addition, none of the studied OCT-A parameters could be used to predict the postoperative VF recovery. 

### 4.1. The Mechanical and Vascular Theories

Although being common, the causes of the visual impairment secondary to CPA have not been fully elucidated. As shown in glaucoma [10,21,22,23], mechanical and vascular disorders could explain the observed functional and anatomical impairments in CPA. Thus, the aim of our study was to determine whether the VD decrease seen in CPA had a primary (vascular theory) or secondary (mechanical theory) origin. 

The mechanical theory postulates that the compression of the optic chiasm by the CPA is associated with a retrograde axonal degeneration leading to the death of retinal ganglion cells, seen as a thinning of the RNFL and GCC on SD-OCT [13]. It has also been hypothesized that the nasal fibers crossing in the optic chiasm were more likely to be damaged by the pressure than the uncrossed temporal fibers, thus explaining traditional bitemporal hemianopsia. Several authors have postulated that a secondary vascular rarefaction could result from the thinning of the retinal GCC, explaining the results of previous studies in which preoperative OCT-A alterations have been reported in CPA [13,16,24]. 

The vascular theory has been widely neglected in the CPA literature. The intracranial part of the optic nerve is mainly vascularized by the ophthalmic artery. Previous studies have reported that the superior hypophyseal artery originates a few millimeters behind the ophthalmic artery (about 5 mm), highlighting a close relationship between these two vascular bundles and a simultaneous compression in case of CPA [2]. Only a few studies have investigated the role of the vascular theory. A few authors have described it in pituitary apoplexy. Pituitary apoplexy is an ischemic hemorrhage occurring within the pituitary gland. Infarction of the inferior hypophyseal arteries is assumed to be the primum movens of the pathology. However, assessing the vascularization of the optic nerve and macula with fluorescein angiography is not recommended in daily clinical practice in case of CPA. The recent development of OCT-A has allowed performing a non-invasive and three-dimensional vascular assessment of the retina and optic nerve. Initially developed for analyzing neovascularization in age-related macular degeneration, OCT-A has been increasingly used to analyze several optic nerve disorders. To assess the role of the vascular theory, we analyzed for the first time preoperative and postoperative OCT-A parameters in CPA patients while making the distinction between patients who experienced a postoperative VF improvement and those who did not. By drawing a parallel with pituitary apoplexy, we could assume that the compression of the superior hypophyseal and ophthalmic arteries by a CPA promote the occurrence of a “chronic ischemic chiasmal syndrome” and a distal macular and papillary VD rarefaction, respectively. We hypothesized that CPA patients with a postoperative VF improvement could show increased papillary and macular VD. It could be assumed that unlike an acute or subacute disease, a chronic compression could lead to an irreversible distal vascular alteration without postoperative VD improvement.

### 4.2. Main Results

The first aim of this study was to assess whether the macular and papillary VD increased after surgery in CPA patients. Ben Ghezala I et al. and Lee G-I et al. have reported a decrease in macular and papillary VD following surgery compared to preoperative values [14,15]. In our study, we arbitrarily defined a VD improvement as a 10% increase in postoperative VD values compared to baseline. Interestingly, using a 5% increase, we found that 14.29% and 16.66% of NCPA and CPA patients experienced a macular sVD improvement, respectively (Table A1). Since there is no evidence to explain the VD improvement seen in the NCPA group, we assumed that OCT-A provided false positive results and concluded that using a 10% increase was more reliable. Taken together, and in accordance with other studies [14,15], our data indicated that the VD was not significantly improved following optic chiasm decompression, suggesting that the vascular theory is not predominant in CPA-associated visual impairment. The altered macular and papillary VD found in CPA patients would be more likely the consequence of a retinal ganglion cell rarefaction due to a retrograde axonal degeneration. Nonetheless, it could not be excluded that a chronic vascular compression could result in an irreversible distal (macular and papillary) vascular rarefaction.

The second aim of our study was to assess whether baseline OCT-A findings could be used to predict the postoperative VF improvement and therefore to adequately inform the patients before surgery. Of interest, the GCC thickness was decreased in the CPA group after surgical decompression (40.97 μm before versus 38.01 μm 6 months after surgery, *p* = 0.03). This could be explained by the time to perform neurosurgery after the ophthalmic examination, during which compression and axonal degeneration progressed. In our study, the RNFL and GCC thicknesses did not correlate with the postoperative VF improvement (Table 4).

Although being the most reliable clinical parameter to assess visual recovery, the preoperative VF was not found to be predictive of the postoperative VF recovery in our multivariate analysis (*p* = 0.07). In addition, the VF is a subjective examination, as evidenced by the fact that we observed an improvement in the VF mean deviation at 1 month in the NCPA group (from −0.52 dB at baseline to −0.24 dB at 1 month, Table 2). Taken together, these findings support the need to identify new reliable preoperative factors for visual outcomes. 

In line with other studies [13,14,19], we found that the macular and papillary VD were lower in the CPA group compared to the NCPA group (Table 3). Once again, the two theories could explain this finding. First, the disappearance of retinal ganglion cells reduces retinal oxygen needs and therefore associated vascular supply (mechanical theory). This could explain why previous studies have shown a correlation between the VD decrease areas and the corresponding VF defect areas. 

Second, the vascular theory supports the fact that a compression of the vascular supply of the optic chiasm and optic nerve could result in a distal macular and papillary VD decrease. In accordance with Ben Ghezala et al., Lee G-I et al. and Parrozzani R et al., we found a slight decrease in postoperative macular sVD in the CPA group [14,17,18]. As previously indicated, several patients in the CPA group also experienced a concomitant decrease in GCC thickness after surgery. Unlike Ga lee in et al., we did not find that the papillary VD and the RNFL and GCC thicknesses were independent prognostic factors for visual recovery in our multivariate analysis (Table 4). Our findings were compared to those from recent comparable studies, as shown in Table A1. 

Taken together, these data suggest that the VD decrease could be secondary to the retinal ganglion cell degeneration (GCC and RNFL thinning), as stated by the mechanical theory.

### 4.3. Study Strengths

First, this is the first prospective controlled study to investigate the benefit of OCT-A in CPA [13,15,16]. Second, unlike other studies that have included healthy subjects as a control group [13,14,15,16], we included NCPA patients who underwent the same surgical treatment as CPA patients, which allowed reducing the confusion bias. For example, we found that several patients in the NCPA group experienced a postoperative VD increase >5% (but none of them experienced an increase >10%), which could be due to an OCT-A calibration bias. Third, we provided a strict definition of VF improvement and dichotomized CPA patients into two groups depending on whether or not their VF was improved after surgery. This allowed better understanding the vascular remodeling and the prognostic factors for VF recovery with exclusion of the most advanced and irreversible cases (chronic compression). We defined the vascular theory, based on previous anatomical and cadaveric studies and on the pathophysiology of pituitary apoplexy. 

### 4.4. Study Limitations

The limitations of this study are its small sample size (19 patients, 38 eyes) and its short follow-up reduced to 6 months. However, we do not think that a longer follow-up would have been useful because none of our patients experienced a persistent optic chiasm compression after surgery. In addition, SD-OCT and OCT-A parameters do not change beyond 3–4 months after surgery [13]. Another bias is the difficulty assessing the exact time of onset of optic chiasm compression. To address this issue, we decided to classify compression duration based on the baseline VF, RNFL and GCC findings. Although not noted when the fundus examination was performed, it cannot be excluded that several patients in the CPA group experienced a subclinical compressive optic disc swelling, which could have affected the VD assessment. However, the postoperative RNFL thickness was not significantly reduced in the CPA group, suggesting that none of our patients had preoperative optic disc swelling. Finally, we chose to analyze the mean macular and papillary VD rather than divide each quadrant (supero-nasal, supero-temporal, infero-nasal, infero-temporal) because it is easier to obtain and to analyze by clinicians in daily clinical practice, and it reduces the risk of false positives provided by the OCT-A device. 

## 5. Conclusions

Our results showed that the surgical decompression of the optic chiasm does not result in a significant improvement in macular and papillary VD, suggesting that the vascular theory is not predominant in CPA-associated VF impairment. In addition, none of the preoperative OCT-A parameters allowed a postoperative VF recovery assessment. Our preliminary results question the use of OCT-A in the routine clinical examination of CPA patients, although larger studies are needed to confirm our findings. 

## Figures and Tables

**Figure 1 diagnostics-14-01747-f001:**
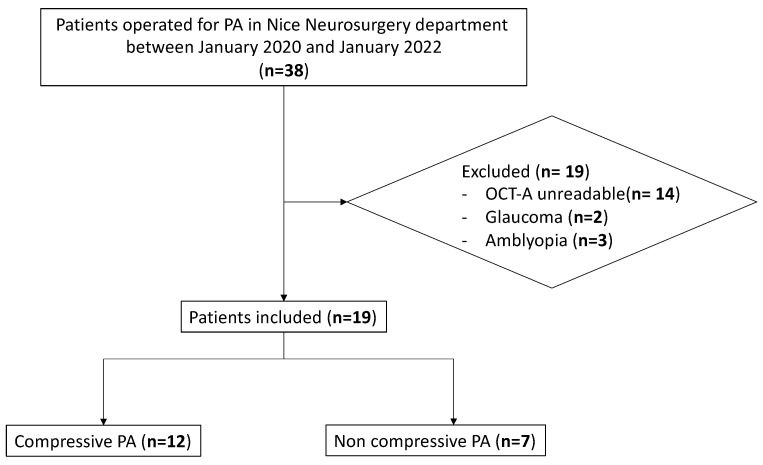
Flow chart.

**Table 1 diagnostics-14-01747-t001:** Baseline demographics and clinical characteristics of patients with compressive and non-compressive pituitary adenoma.

Patients’ Characteristics		All Groups (*n* = 19)	CPA (*n* = 12)	NCPA (*n* = 7)	*p **
Number of eyes		38	24	14	0.39
Age (years), mean (range) [SD]		55 (33–78) [14]	59 (33–78) [13]	50 (36–76) [15]	0.17
Gender, *n* (%)	Male	6 (31.6)	6 (50)	0	**0.04**
	Female	13 (68.4)	6 (50)	7 (100)	
Secreting PA, n (%)		8 (42.1)	2 (16.7)	6 (66.7)	**<0.01**
Duration of optic chiasm compression, *n* (%)	Acute	NA	1 (8.3)	NA	
	Subacute	NA	6 (50)	NA	
	Chronic	NA	5 (41.7)	NA	
Hormone replacement therapy, *n* (%)		8 (42.1)	2 (16.7)	6 (66.7)	**<0.01**
Diabetes, *n* (%)		1 (5.3)	0	1 (14.29)	0.37
High blood pressure, *n* (%)		4 (21.1)	2 (16.67)	2 (28.57)	0.60
Smoking, *n* (%)		0	0	0	1.00
Cataract, *n* (%)		4 (21.1)	3 (25.00)	1 (14.29)	1.00
IOP (mmHg), mean (range) [SD]		15.60 (13–21) [2.34]	15.30 (13–20) [1.85]	16.10 (13–21) [3.02]	0.64
Disc pallor, *n* (%)		5 (19)	5 (41.67)	0	0.11
BCVA (LogMAR), mean (range) [SD]		0.13 (0.00–0.07) [0.23]	0.20 (0.00–0.07) [0.28]	0.02 (0.00–0.02) [0.04]	0.06
VF mean deviation (dB), mean (range) [SD]		−6.09 (−20.18–1.1) [6.40]	−9.34 (−20.18–0.28) [5.96]	−0.52 (−1.79–1.10) [0.86]	**<0.01**
RNFL thickness (μm), mean (range) [SD]		97.69 (59.50–115.25) [15.22]	91.13 (59.50–115.25) [15.61]	108.93 (95.84–114.75) [3.50]	**<0.01**
GCC thickness (μm), mean (range) [SD]		43.96 (20.50–57.25) [8.76]	40.97 (20.50–57.25) [9.63]	49.09 (42.76–53.75) [3.01]	**0.02**
Whole disc VD (%), mean (range) [SD]		53.18 (43.70–59.30) [3.74]	51.50 (43.70–59.30) [3.49]	55.90 (50.08–59.30) [2.23]	**<0.01**
Inside disc VD (%), mean (range) [SD]		59.23 (47.50–65.60) [3.81]	59.07 (47.50–64.00) [3.59]	59.61 (51.70–65.60) [4.16]	0.72
Disc peripapillary VD (%), mean (range) [SD]		54.61 (41.30–64.70) [5.05]	52.15 (41.30–56.80) [4.47]	58.83 (55.10–64.70) [2.59]	**<0.01**
Macular parafoveal sVD (%), mean (range) [SD]		49.01 (39.00–82.25) [6.77]	48.66 (39.00–82.25) [8.29]	49.61 (45.00–53.50) [2.91]	0.33
Macular parafoveal dVD (%), mean (range) [SD]		51.39 (26.25–64.25) [8.25]	51.73 (36.50–61.50) [7.48]	50.83 (26.25–64.25) [9.73]	0.79
Foveal avascular zone (%), mean (range) [SD]		22.32 (6.00–82.00) [12.24]	22.54 (6.00–82.00) [14.90]	19.36 (12.00–26.00) [4.33]	0.33
Pituitary adenoma volume (cm^3^), mean (range) [SD]		8.11 (0.16–13.20) [1.51]	10.65 (5.30–13.20) [1.91]	3.77 (0.16–8.50) [1.48]	**<0.01**
Surgery duration (min), mean (range) [SD]		122.2 (74.00–390.00) [68.46]	137.4 (74.00–390.00) [82.61]	98.8 (79.00–125.00) [16.84]	0.06
Intraoperative complication, *n* (%)		1 (5.27)	1 (8.33)	0 (0)	1.00
Ki67 (%), mean (range) [SD]		1.68 (0–10.00) [2.56]	1.17 (0–5.00) [1.85]	2.57 (0–10.00) [3.46]	0.27
Cell mitosis (% per 10 field), mean (range) [SD]		0.68 (0.00–6.00) [1.45]	0.33 (0.00–6.00) [0.65]	1.29 (0.00–2.00) [2.21]	0.36

BCVA, best-corrected visual acuity; CPA, compressive pituitary adenoma; NCPA, non-compressive pituitary adenoma; PA, pituitary adenoma; IOP, intraocular pressure; VF, visual field; RNFL, retinal nerve fiber layer; GCC, ganglion cell complex; sVD, superficial vascular density; dVD, deep vascular density; SD: standard deviation. (*) statistical analyses were performed between the CPA group and the NCPA group.

**Table 2 diagnostics-14-01747-t002:** Main visual outcomes in the NCPA and CPA groups.

			NCPA			CPA		
	M0	M1	M6	*p **	M0	M1	M6	*p **
Mean BCVA (LogMAR)	0.02	0.01	0.02	1.00	0.2	0.03	0.04	**0.02**
VF Mean Deviation (dB), mean	−0.52	−0.24	−0.60	0.85	−9.34	−3.35	−2.01	**<0.01**
VF Improvement, *n* (%)	NA	0	0	NA	NA	4 (33)	6 (50)	NA
Mean RNFL Thickness (μm)	108.93	110.48	107.92	0.56	91.13	91.23	89.26	0.30
Mean GCC Thickness (μm)	49.09	48.27	49.23	1.00	40.97	39.28	38.01	**0.03**
Mean Tumor Size (cm^3^)	3.17	NA	0.13	**0.02**	8.84	NA	3.50	**<0.01**

CPA, compressive pituitary adenoma; NCPA, non-compressive pituitary adenoma; BCVA, best-corrected visual acuity; VF, visual field; VF improvement, presence of a mean deviation improvement >6 dB and subjective VF defect improvement; RNFL, retinal nerve fiber layer; GCC, ganglion cell complex; NA, not available. (*) comparison between M0 and M6.

**Table 3 diagnostics-14-01747-t003:** Changes in OCT-A parameters 6 months after surgery compared to baseline.

		Mean Macular sVD (%)				Mean Whole Disc VD (%)				Mean Inside Disc VD (%)				Mean Disc ppVD (%)			
		M0	M6	i ≥ 10%, n	VD Change (%)	M0	M6	i ≥ 10%, n	VD Change (%)	M0	M6	i ≥ 10%, n	VD Change (%)	M0	M6	i ≥ 10%, n	VD Change (%)
NCPA		49.61	48.95	0	−1.33	55.9	56.44	0	0.97	59.61	59.51	0	−0.17	58.83	58.86	0	0.05
CPA	total	48.66	**46.18 ***	1	−5.10	51.5	51.16	0	−0.66	59.07	60.03	0	1.63	52.15	51.48	0	−1.28
	with VF improvement	47.63	47.02	1	−1.28	52.81	52.60	0	−0.40	58.02	59.25	0	2.12	53.86	53.46	0	−0.74
	without VF improvement	49.71	45.33	0	−9.66	50.19	49.73	0	−0.92	60.12	60.82	0	1.16	50.43	49.48	0	−1.88

*n*, number; NCPA, non-compressive pituitary adenoma; CPA, compressive pituitary adenoma; VF, visual field; sVD, superficial vascular density; VD, vascular density; ppVD, peripapillary vascular density; i ≥ 10%, improvement ≥ 10%. (*****) statistically significant.

**Table 4 diagnostics-14-01747-t004:** Univariate and multivariate analyses of preoperative data related to the postoperative visual field improvement.

Variable	Univariate Analysis		Multivariate Analysis	
	OR, 95% CI	*p*-Value	OR, 95% CI	*p*-Value
Age		0.49		
Gender	**11 (1.14–106.43)**	**0.04**	0.76 (0.56–1.01)	0.06
Secreting PA		0.15		
Diabetes		1.00		
High blood pressure		0.75		
Smoking		1.00		
IOP		0.55		
Disc pallor		0.13		
BCVA		0.90		
VF mean deviation	**0.79 (0.63–0.98)**	**0.04**	15.81 (0.76–327.08)	0.07
Duration of optic chiasm compression		0.30		
RNFL thickness		0.81		
GCC thickness		0.61		
Whole disc VD		0.78		
Inside disc VD		0.23		
Disc peripapillary VD		0.63		
Macular parafoveal sVD		0.43		
Macular parafoveal dVD		0.82		
Foveal avascular zone		0.90		
Surgery duration		0.51		

VF improvement, improvement in median deviation > 6 dB and improvement in subjective VF at M6; BCVA, best-corrected visual acuity; PA, pituitary adenoma; IOP, intraocular pressure; VF, visual field; RNFL, retinal nerve fiber layer; GCC, ganglion cell complex; VD, vascular density; sVD, superficial vascular density; dVD, deep vascular density; CI, confidence interval; OR, odds ratio.

## Data Availability

The original contributions presented in the study are included in the article, further inquiries can be directed to the corresponding author.

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
