# Peer review of "Benefit of Optical Coherence Tomography–Angiography in Patients Undergoing Transsphenoidal Pituitary Adenoma Surgery: A Prospective Controlled Study"

_diagnostics, 2024, doi:10.3390/diagnostics14161747_

Round 1

Reviewer 1 Report

Comments and Suggestions for Authors

1.       Original Submission

Recommendation to the author and editor:

Major revision

Title: Manuscript ID: diagnostics-3115610 entitled " Benefit of Optical Coherence Tomography-Angiography in 2 Patients Undergoing Transsphenoidal Pituitary Adenoma 3 Surgery: A Prospective Controlled Study".

Article Type: Original Paper

2.       Comments to the Corresponding Author:

COPE Ethical guidelines followed during the review process,

The manuscript addresses that this study was to determine whether visual density improved after surgical decompression of the optic chiasm in CPA patients and whether OCT-A could help identify predictive factors for postoperative visual recovery. Authors performed this clinical trial ID: NCT04074642 as a prospective controlled study with patients undergoing transsphenoidal pituitary adenoma surgery, divided into those with CPA and those without CPA (NCPA). Authors performed neuro-ophthalmological examinations, visual field (VF) testing, macular and optic disc structural OCT (retinal nerve fiber layer (RNFL) and ganglion cell complex (GCC) thicknesses), and OCT-A before surgery, and then at 1 and 6 months post-surgery for all the registered patients in this trial.

According to authors, the study included 24 eyes from the CPA group and 14 eyes from the NCPA group. None of the VD parameters assessed by OCT-A showed significant improvement after surgery in the CPA group; in fact, the mean macular superficial VD significantly decreased at 6 months. Authors performed multivariate analysis but which did not identify any preoperative parameters predictive of postoperative VF improvement. Our preliminary findings suggest that the visual impairment observed in CPA patients could not be explained by the vascular theory, and none of the preoperative OCT-A parameters allowed for an assessment of postoperative VF recovery. As per my view, Author’s future research should focus on longer follow-up periods, increased sample sizes, advanced imaging techniques, exploration of alternative mechanisms of visual impairment, identification of reliable predictive biomarkers, and the development of personalized treatment plans to optimize visual recovery in CPA patients.

Comments:

Overview and general recommendation:

The paper was well written. Yet, proofreading can enhance the quality of the manuscript. Several sentences need rewriting to make the readers comfortable when reading this. Avoid spelling errors.

1.               Authors should expand the introduction and discussion parts with additional content by studying variable reports comparatively with current clinical outcome. However, patients’ sample is literally low to conclude ‘preoperative OCT-A parameters allowed a postoperative VF recovery assessment.’

2.               Expand the Figure legends with vivid explanation.

3. Conclusion should be explained vividly.

4. I am satisfied with data and follow up if the patients. Change ‘preoperative oct a’ into ‘preoperative OCT-A’.

                                                **Thank you**

Comments on the Quality of English Language

Can be improved. Follow ENdnote style of referencing. 

Author Response

Thank you very much for your comments.

  1. Authors should expand the introduction and discussion parts with additional content by studying variable reports comparatively with current clinical outcome. However, patients’ sample is literally low to conclude ‘preoperative OCT-A parameters allowed a postoperative VF recovery assessment.’ : we explaned more vivdly the discussion part. We agree with your comment and the conclusion of the study is that NONE of the preoperative OCT-A parameters allowed a postoperative VF recovery assessment, due to, partly, the small sample of patients.
  2. Expand the Figure legends with vivid explanation : the figure are interpreted in the manuscript (result and discussion) where the explanations are made. We prefere to keep the legend short and concise and explain the figure in the main text of the manuscript.
  3. Conclusion should be explained vividly : we explained more vividly our conclusion in the discussion part, and kept the conclusion short and simple for more quality.
  4. I am satisfied with data and follow up if the patients. Change ‘preoperative oct a’ into ‘preoperative OCT-A’ : this change has been made.

Reviewer 2 Report

Comments and Suggestions for Authors

Good study, well presented.

Author Response

Thank you very much for your comments.

Comments and Suggestions for Authors

Good study, well presented.

Reviewer 3 Report

Comments and Suggestions for Authors

This paper conducted a prospective controlled study in 22 patients who underwent transsphenoidal pituitary adenoma surgery with two groups: CPA and NCPA to determine whether the VD improved after surgical decompression of the optic chiasm in CPA patients and whether OCT-A could help to identify predictive fac-21 tors for postoperative visual recovery.

There are some problems to be solved before publication.

(1)  What’s the logical relationship between Transsphenoidal Pituitary Adenoma and the CPA. It is not clear in the introduction section.

(2)  Some abbreviations in this paper should be given the full names such as VD in abstract and SD-OCT in line 60.

(3)  The font could be large for read in Figure 1.

Author Response

Thank you very much for your comments.

(1)  What’s the logical relationship between Transsphenoidal Pituitary Adenoma and the CPA. It is not clear in the introduction section : Compressive pituitary adenoma (CPA) can be operated by transsphenoidal  endoscopic surgery. It is the neurosurgical technique that was used in the study. A transsphenoidal pituitary adenoma does not exist. 

(2)  Some abbreviations in this paper should be given the full names such as VD in abstract and SD-OCT in line 60 : indeed, we have written the full name.

(3)  The font could be large for read in Figure 1 : we have enlarged the policy.